# Exploring *Pijuayo* (*Bactris gasipaes*) Pulp and Peel Flours as Fat Replacers in Burgers: A Multivariate Study on Physicochemical and Sensory Traits

**DOI:** 10.3390/foods13111619

**Published:** 2024-05-23

**Authors:** Alex Y. Llatas, Heiner Guzmán, Fernando Tello, Roger Ruiz, Jessy Vásquez, Grisel Chiroque, Jhony Mayta-Hancco, Melina L. M. Cruzado-Bravo, Hubert Arteaga, Erick Saldaña, Juan D. Rios-Mera

**Affiliations:** 1Instituto de Investigación de Ciencia y Tecnología de Alimentos (ICTA), Universidad Nacional de Jaén, Jaén 06800, Peru; alex.llatas@est.unj.edu.pe (A.Y.L.); heiner.guzman@est.unj.edu.pe (H.G.); hubert.arteaga@unj.edu.pe (H.A.); juan.rios@unj.edu.pe (J.D.R.-M.); 2Departamento de Ingeniería de Alimentos, Facultad de Industrias Alimentarias, Universidad Nacional de la Amazonía Peruana, Iquitos 16002, Peru; roger.ruiz@unapiquitos.edu.pe (R.R.); jessy.vasquez@unapiquitos.edu.pe (J.V.); 3Escuela Profesional Industrias Alimentarias, Facultad de Ingeniería, Universidad Nacional de Barranca, Av. Toribio Luzuriaga Urb. La Florida 376, Barranca 150201, Peru; gchiroque@unab.edu.pe; 4Sensory Analysis and Consumer Study Group, Escuela Profesional de Ingeniería Agroindustrial, Universidad Nacional de Moquegua, Moquegua 18001, Peru; jmaytah@unam.edu.pe (J.M.-H.); esaldanav@unam.edu.pe (E.S.); 5Grupo de Investigación en Desarrollo, Calidad y Seguridad de Alimentos (GIDCSA), Escuela de Ingeniería Agroindustrial, Facultad de Ciencias Agrarias, Universidad Nacional Autónoma de Chota, Chota 06120, Peru; mlcruzadob@unach.edu.pe

**Keywords:** Amazon fruits, fat substitutes, physicochemical characteristics, sensory characteristics, multivariate statistical analysis

## Abstract

Meat products are known for their lipid profile rich in saturated fats and cholesterol, and also for the formation of oxidation compounds; therefore, a reduction in animal fat may result in a product less harmful to health. *Pijuayo* is an Amazon fruit known for its nutritional properties, such as its fiber and lipid content. For these reasons, it is an attractive fruit to replace animal fat in meat products. The present work used *pijuayo* pulp and peel flours to partially replace animal fat in beef-based burgers at 25% and 50% levels, considering sensory and physicochemical outcomes evaluated by Principal Component Analysis (PCA), Correspondence Analysis (CA) and Multiple Factor Analysis (MFA). *Pijuayo* flour affected the physicochemical characteristics evaluated by PCA, where the samples with greater fat replacement were characterized by a high carbohydrate content and instrumental yellowness. The minimal fat replacement did not abruptly affect the PCA’s instrumental texture and color, proximal composition, yield properties, and lipid oxidation. The overall liking was greater in the 25% fat reduction treatments, even greater than the control, in which positive sensory attributes for liking were highlighted for those treatments. A small segment of consumers (11% of total consumers) preferred the treatment with greater replacement of fat with *pijuayo* peel flour, which these consumers tended to characterize as *seasoned*. However, this treatment had the lowest liking. The MFA showed that the sensory characteristics tender and tasty were strongly correlated with overall liking and were highlighted in the samples of 25% fat reduction, suggesting that the *pijuayo* improves the tenderness and flavor of reduced-fat burgers. Other inclusion levels between 25% and 50% of fat replacement could be explored, and optimization studies are needed. In addition, the sensory characteristics and flavor-enhancing compounds of the fruit, as well as the nutritional aspects of the inclusion of *pijuayo,* should be studied, such as the fatty acid profile. These characteristics will be informative to explore *pijuayo* as a fat replacer at a pilot scale and industrial scale.

## 1. Introduction

Burgers are popular among consumers because of their sensory appeal. However, they are high in saturated fats and cholesterol due to the use of animal fats in their preparation. The Mediterranean diet, known for its health benefits, has been shown to protect against cardiovascular risks associated with consuming processed meats with unhealthy lipid profiles [1]. Thus, using plant-based alternatives in burger formulations is a promising strategy [2]. This approach particularly appeals to consumers who find it difficult to change their dietary habits due to cultural and economic factors associated with meat products.

Several botanical resources are available to replace animal fat in meat products, such as vegetable flours and healthy oils [3]. Although the Peruvian Amazon is a biome with vegetable resources that can be used in the food industry and have widely reported health benefits [4,5,6], little interest has been shown in this matter. In burgers, freeze-dried hydrogels containing açai oil were used to substitute animal fat, which improved the lipid profile and reduced cooking losses. However, it was also noted that lipid oxidation tended to increase [7]. Echeverria et al. [8] used flour from the stem of peach palm (*Bactris gasipaes*) to partially replace pork fat in lamb burgers. The cooking yield, moisture retention, and dietary fiber content increased, but the sensory attributes were acceptable for substitution up to 6% of the stem of *Bactris gasipaes* flour. These studies suggest that technological defects can be overcome with adequate fat substitution levels or using antioxidants. However, the manufacturing process of the fat substitute should also be considered, as it should not significantly affect the production cost of the reduced-fat product compared to the conventional product. Thus, obtaining vegetal flour to replace animal fat can be considered a simpler and cheaper alternative.

Recent research suggests substituting animal fat with fruit flour in burgers can produce promising technological results. These fat-substitute flours are typically rich in fiber and contain compounds that can help prevent lipid oxidation in burgers [8,9,10,11,12]. One fruit that meets these criteria exceptionally well is *Bactris gasipaes*, also known as “pijuayo” in Peru. *Pijuayo* is a popular fruit among Amazonian settlers. It is integrated into agroforestry systems, in gardens, orchards, and in commercial plantations to produce heart of palm. Each tree produces 5–10 bunches of fruits per year, and up to 12 kg of fruits per year [6]. It has high levels of phenolic compounds, carotenoids, fiber, essential amino acids, and unsaturated lipids, which possess antioxidant, antimicrobial, and anti-obesogenic properties [4,6]. As such, *pijuayo* shows excellent potential as a fat replacement in burgers.

Recently, we published a proceeding paper on the effect of *pijuayo* pulp and peel flours on the physicochemical characteristics of beef-based burgers, with positive results in yield properties, fat reduction, and lipid oxidation. However, the texture profile parameters were significantly reduced by including *pijuayo* flour [13]. To determine if these changes are positive, it is necessary to complement it with sensory properties and acceptance of the product. This information will allow us to define adequate levels of fat replacement, as well as to explore the sensory attributes affected by the inclusion of *pijuayo* flour that explain the consumers liking. In this context, challenges could be identified to continue exploring the potential industrial application of *pijuayo* flour as a fat substitute.

On the other hand, it is often difficult to recommend an appropriate level of reduction/replacement of fat in meat products, as the effects on instrumental and/or sensory traits can be positive, negative, or inconclusive when viewed individually. To better understand the effect of reformulation, it is helpful to visualize instrumental and sensory data from a multivariate perspective, which can provide an overview of the factors associated with product liking. In this regard, Rios-Mera et al. [14] studied this approach using Multiple Factor Analysis (MFA) and found that the acceptance of salt-reduced burgers was strongly correlated with sensory attributes such as juicy, tasty, salty, and seasoned, as well as with instrumental hardness and chewiness between 78.68 and 96.11 Newtons (N) and 42.28 and 47.96 N, respectively.

In this framework, the aim of this study was to investigate how substituting animal fat (pork backfat) with *pijuayo* pulp and peel flours affects the physicochemical and sensory characteristics of beef-based burgers from a multivariate perspective, using Principal Component Analysis (PCA), Correspondence Analysis (CA), and MFA. Before MFA, an internal preference map and dendrogram were also constructed to identify clusters or consumer segments associated with their preferences in the sensory evaluation. Thus, it is important to conclude whether the association of instrumental and sensory characteristics with acceptance in the MFA is representative of consumers who evaluate the product.

## 2. Materials and Methods

### 2.1. Materials

Lean beef, pork backfat, table salt, and monosodium glutamate were purchased from supermarkets in Jaén, Cajamarca, Peru. Onion, garlic, and pepper powder were purchased from Alitecno (Lima, Peru). Sodium erythorbate was purchased from Frutarom Perú S. A. (Lima, Peru). The fruits of *pijuayo* were collected in a ripe state, defined by the orange-red color, from the Aguas Verdes sector (1234 m above mean sea level) of the Pardo Miguel district, province of Rioja, San Martín, Peru. The fruits were transported in hermetic bags to the Laboratory of Food Technology of the National University of Jaén, where they were processed, taking an approximate transport time of 5 h.

### 2.2. Pijuayo Pulp and Peel Flour Processing and Characterization

The fruits were first washed with water and then cooked in boiling water for 30 min. After that, the pulp and peel were separated using a knife. The flours were obtained by dehydrating the pulp and peel in an oven with air circulation at 55 °C (Kertlab, model ODHG-9053A, USA) until the moisture level was less than 15%. Once this was achieved, the dehydrated pulp and peel were ground with a domestic mill and sieved (250 µm). Finally, the flours were stored under vacuum at −18 °C until analysis and burger processing.

The *pijuayo* pulp and peel flours were characterized in their proximal composition and the instrumental color parameters L* (lightness), a* (redness), and b* (yellowness) in triplicate. For proximal composition, the AOAC [15] methods were used. For moisture, the thermogravimetric method in an electric oven (JP SELECTA, Barcelona, Spain) at 105 °C for 5 h was used (method 950.46); ash was determined by incineration of samples in a muffle (Furnace 1400, Thermo Scientific, Waltham, MA, USA) at 550 °C for 6 h (method 920.153); protein was performed using the Kjeldahl method in a digester and distiller (Behrotest K8-SR3, Düsseldorf, Germany). The calculation was based on nitrogen content considering 6.25 as conversion to protein (method 981.10), and fat was determined by the Soxhlet method, using petroleum ether as solvent (method 960.39). Carbohydrate was calculated by difference using Equation (1):(1)Carbohydrate (g/100 g)=100−(Moisture+Ash+Protein+Fat) 

The color parameters were obtained through a computer vision system. This system uses a smartphone to capture the image, which is then processed by the IP-WEBCAM-COLOR FOOD PROCESSING algorithm implemented in MatLab® R2016a [16]. Figure 1 shows the *pijuayo* processing scheme and the characteristics of the flours.

### 2.3. Burger Manufacture

The burger manufacturing process was the same as that in Guzmán et al. [13], which included the following ingredients: lean beef (70%), pork fat (from 20% to 10%, according to the treatment), *pijuayo* pulp or peel flour (from 5% to 10%, according to the treatment), cold water (7.5%), monosodium glutamate (0.28%), garlic powder (0.28%), onion powder (0.28%), white pepper powder (0.15%), and sodium erythorbate (0.01%). The study included five treatments, which were manufactured as follows: T1—0% pork fat substitution or control; T2—25% pork fat substitution with *pijuayo* pulp flour; T3—50% pork fat substitution with *pijuayo* pulp flour; T4—25% pork fat substitution with *pijuayo* peel flour; and T5—50% pork fat substitution with *pijuayo* peel flour. The processing of the burgers was repeated in three independent batches. In each batch, 30 burgers were processed for each treatment. Representative burgers from each treatment are shown in Figure 2, where differences in color due to the inclusion of *pijuayo* flour can clearly be observed. Thus, the instrumental color results of the samples are also included in Figure 2.

The ingredients were mixed by hand for 5 min. Then, 100 g portions of burger batter were shaped into burgers using a 10 cm diameter and 1 cm thick mold. Next, three samples of each treatment per batch were used for instrumental color measurement and pH. The burgers were then stored at −18 °C for about 2 h to prevent any deformation during vacuum packaging. The vacuum-sealed burgers were then stored at −18 °C for a maximum of two weeks.

For sensory analysis, the samples were cooked on an electric hot plate at 150 °C until the internal temperature of the burgers reached 75 °C. Then, the samples were cooled to 45 °C.

### 2.4. Physicochemical Characteristic Data

Data of the proximal composition, texture profile analysis, cooking losses, diameter reduction, and lipid oxidation (thiobarbituric acid reactive substance—TBARS) of burgers were extracted from our previous report [13], to express the results as PCA, detailed in the data analysis section. Additionally, the instrumental color (L*, a*, b*) was determined as detailed in Section 2.2, as well as the pH of the raw burgers, measured in three different points of burger samples with a pH-meter (Metrohm, model CH-9100, Herisau, Switzerland) coupled to a glass electrode, previously calibrated in buffer solutions of pH 4, 7 and 10.

### 2.5. Sensory Analysis

#### 2.5.1. Consumers

The consumer panel was made up of 112 regular burger consumers, as per the recommendation of Hough et al. [17]. The panel consisted of 68% women, 31% men, and 1% who preferred not to declare their sex. The participants were aged between 18 and 58 years old. They reported consuming burgers at different frequencies: 3% ate burgers six to four times a week, 10% ate burgers three to one times a week, 18% ate burgers every 15 days, 30% ate burgers once a week, and 39% rarely ate burgers. Before participating in the study, consumers had to read and sign a consent form to approve their participation.

#### 2.5.2. Procedure

Before sensory analysis, the burgers were microbiologically analyzed (aerobic mesophilic, Staphylococcus aureus, and Escherichia coli) to ensure the eating quality, according to Downes and Itō [18]. In this regard, the microorganism count was between 2.3 and 3.2 × 106 CFU/g aerobic mesophilic, <10 CFU/g Staphylococcus aureus, and between 1.1 and 4.2 × 10 CFU/g Escherichia coli, which were within the limits according to the Peruvian regulations (107, 103 and 5 × 102, respectively) [19].

The sensory analysis was conducted in individual sensory booths in a single session. The Check-All-That-Apply (CATA) method was used to describe the burger samples using sensory terms extracted from Rios-Mera et al. [14]. These terms were obtained from consumers’ vocabulary who characterized burgers in a previous study [20]. Samples of approximately 10 g were served monadically and balanced on plates coded with three-digit random numbers using a Williams Latin Square design [21]. The participants rated their overall liking using a 9-point hedonic scale, ranging from “extremely dislike” to “extremely like” [22], and then were asked to select all sensory terms that applied to describe the samples. Water and unsalted crackers were provided to consumers to cleanse the palate between samples.

### 2.6. Data Analysis

Data on burger physicochemical characteristics were analyzed by PCA, considering the average values of each characteristic for each treatment.

A mixed ANOVA was used for overall liking, with treatment as a fixed factor and consumer and presentation order as random factors. Tukey’s test was used for treatment pairwise comparison at 5% significance. An internal preference map and dendrogram were also created to represent consumer preference into clusters [23].

The CATA data were analyzed by calculating each sample’s elicitation frequencies for sensory terms, called the contingency table. Then, a Correspondence Analysis (CA) on the contingency table was performed to represent the samples and attributes multivariably. A Penalty Analysis (PA) was also conducted to determine the impact of sensory characteristics on the burgers’ overall liking.

An MFA was used to correlate physicochemical and sensory data (arranged as active tables), considering the treatments as individuals, and the mean overall liking was included as a supplementary variable [14]. Only the physicochemical variables that differentiated the treatments in the PCA and the sensory characteristics that impacted overall liking for more than 20% of consumers in the PA were included.

XLSTAT 2015 (Addinsoft, New York, NY, USA, EEUU) and R version 4.3.1 software [24] were used for data analysis.

## 3. Results

### 3.1. Principal Component Analysis (PCA) of Physicochemical Characteristics

The PCA of the physicochemical characteristics is shown in Figure 3. The first two principal components represent 82.7% of the data variability, and the differences between the treatments are notable in the first principal component, where three groups can be observed: burgers with the higher fat replacement (50%) (T3 and T5), burgers with the lower fat replacement (25%) (T2 and T4) and the control (T1, without fat replacement); that is, the substitution of animal fat using *pijuayo* flour affected the physicochemical characteristics of burgers from a multivariate point of view.

The greatest associations were for T1, standing out as the product with the greatest hardness, chewiness, cohesiveness, springiness, ash, lipid oxidation (TBARS), and redness (a*), and the lowest cooking yield. At the other extreme of the first component of the PCA were the treatments with the higher inclusion of *pijuayo* flour (T3 and T5), meaning they obtained the lowest associations with the parameters highlighted in T1. These treatments were associated with carbohydrates and yellowness (b*) because the *pijuayo* flours obtained high values of these characteristics (Figure 1). Guzmán et al. [13] did not observe significant differences for carbohydrates, and for ash, there was no clear trend regarding the inclusion of *pijuayo* flour. Thus, the PCA helps to better observe the trends of the treatments with their characteristics. The treatments with less fat replacement (T2 and T4) occupied an intermediate position in the PCA between the control and the treatments with greater fat replacement; that is, the changes in the physicochemical parameters were not abrupt. Protein, moisture, L*, and pH (pH was in the range of 6.00 and 6.02 for the treatments) were not determinants to differentiate the samples. However, Guzmán et al. [13] observed a slight increase in the moisture of burgers with *pijuayo* flours, around 2 to 3% increase, from the perspective of univariate statistical analysis.

A reduction in fat content is expected when *pijuayo* flour is used as a fat substitute. The ash content of the product was similar to that reported by Selani et al. [11], who suggested that the reduction in ash in low-fat burgers is probably due to the replacement of fat with water. However, in this study, the water added remained the same for all treatments. Therefore, the decrease in ash content was likely due to the low ash content of the *pijuayo* flours (Figure 1).

Replacing fat with *pijuayo* flour decreased the texture characteristics evaluated, including hardness, springiness, cohesiveness, and chewiness. However, studies by Selani et al. [11] and Echevarria et al. [8] observed an increase in hardness and chewiness, with little to no effect on springiness and cohesiveness. There could be several explanations for these differences, such as the product’s water content, the fats’ role in tenderness, the quality of the fiber of the fat substitute flour, and the flour’s interference in forming protein gel that impacts texture. This study hypothesizes that the decrease in texture parameters is mainly due to the improved retention of components such as water and fat in the treatments with *pijuayo* flours, which is expressed by the lower associations with cooking losses and diameter reduction in the PCA (Figure 3).

In addition to the differences in b* between the control and low-fat treatments, the low-fat burgers had a low association with a* due to the low a* in the *pijuayo* flours (Figure 1). Interestingly, the *pijuayo* peel’s external part had an orange-red color (Figure 1c), which explains the highest value of a* compared to the pulp flour, but the yellowness was similar to that of the pulp because the internal part of the peel stood out in yellow color (Figure 1), which resulted in similar b* values for treatments with the same level of inclusion of pulp and peel flours (Figure 2).

The addition of *pijuayo* flours was found to decrease lipid oxidation (TBARS values). This may be due to the presence of phenolic compounds or carotenoids with antioxidant activity [4,6,25]. However, further studies are required to determine the specific contribution of different compounds found in the pulp and peel of *pijuayo* on the oxidative stability of reduced-fat burgers. This would enable a more precise understanding of how *pijuayo* pulp and peel can be used to substitute fat in burgers, maintaining their oxidative stability.

Therefore, the results of the PCA show complementary information to the previous work of Guzmán et al. [13], mainly in the color parameters, ash, carbohydrate, and moisture of the burgers. The magnitude of the physicochemical changes should be associated with the sensory properties and acceptance of the product, to recommend precisely which *pijuayo* by-product (pulp or peel) and level of fat replacement should be applied.

### 3.2. Sensory Analysis

The mean overall liking was in the range of 6.33 to 7.08 for all treatments (Figure 4). Burgers containing 25% less animal fat received the highest overall liking scores even more than the control treatment. Studies of meat products substituted in fat with sensory scores higher than the control are uncommon. Patinho et al. [26] observed a similar trend in their study, when they replaced up to 75% of pork fat content in burgers with *Agaricus bisporus* mushroom. The increases in acceptability are attributed to the characteristics of the fat substitutes. In the case of mushrooms, the presence of umami compounds (aspartic acid, glutamic acid and 5′-nucleotides) improves the flavor of foods [26], while Costa et al. [27] indicate that the *pijuayo* fruit with oily characteristics is highly appreciated by consumers. Amorim et al. [28] reported that the lipid content of the *pijuayo* fruit varies from 8 to 23%, with a prevalence of unsaturated fatty acids. In this work, the average fat contents for the *pijuayo* pulp and peel flours were 10.19 and 11.44 g/100 g, respectively (Figure 1), values that were not low enough to cause detrimental effects to the acceptability of the burgers due to the partial removal of animal fat. Furthermore, *pijuayo* fruit is an important source of essential and non-essential amino acids, and in the latter glutamic acid and aspartic acid stand out [6,29]. On the other hand, Teixeira et al. [30] reported that the *pijuayo* fruit is sensorially characterized as “sweet” and “starchy”. Therefore, it is likely that the composition and sensory characteristics of the *pijuayo* fruit were relevant in improving the acceptance of the burgers. However, in this study, it is not possible to make associations between the composition or sensory characteristics of the *pijuayo* fruit or its flours with the burgers containing *pijuayo* flour, but we strongly suggest exploring flavor-enhancing compounds as well as sensory characteristics of the fat substitute before incorporating it into the reduced-fat meat product, an approach that has not been explored and that can complement the technological and sensory characteristics to explain in depth the acceptance of the reformulated products.

Another characteristic associated with acceptance can be related to the texture of the fat-reduced burgers. In this context, *pijuayo* flour increased the cooking yield and reduced the texture parameters of the burgers. Similar to our results, Patinho et al. [26] observed that *Agaricus bisporus* mushroom decreased hardness, chewiness, cooking losses, and diameter reduction of burgers with 50% animal fat replacement, in which the acceptability was significantly higher than the control.

The sensory characteristics of the burgers were explored with CA and the impact of the characteristics on overall liking were determined with PA. In the CA explained by 93.11% of the first two dimensions (Figure 5a), the treatment with characteristics similar to the control was the one with the lower inclusion of *pijuayo* pulp flour (T2), both characterized by seasoned, characteristic, juicy, aromatic, and beef. Then, there was the treatment with the lower inclusion of *pijuayo* peel flour, mainly characterized as tender and tasty. Next, the burger with the greatest inclusion of *pijuayo* pulp flour (T3) did not have outstanding sensory characteristics because it was located close to the line that divides the upper quadrants without proximity to sensory characteristics. Finally, the treatment with greater inclusion of *pijuayo* peel was related to unfavorable sensory characteristics, such as dry, spicy, and compact.

The PA revealed that the attributes tasty, tender, aromatic, juicy, and seasoned positively increased the overall liking of burgers for more than 20% of consumers (Figure 5b). Other attributes that were also positive but with a lesser impact were beef, grilled, and characteristic. All the positive attributes except grilled characterized T1, T2, and T4 in the CA, while dry was the attribute that negatively impacted overall liking and was characteristic of T5 (Figure 5a,b). The present study found that most drivers of liking aligned with those identified in a previous study by Rios-Mera et al. [14], who used the same descriptors. However, some differences may be due to the experiment’s nature and the ingredients used.

While the association of burgers with 25% fat replacement by *pijuayo* flour with the positive sensory drivers of liking as *tender* and *juicy* can be explained by the yield properties and instrumental texture, the associations with *tasty*, *aromatic*, and *seasoned* further reinforce the idea of exploring flavor-enhancing compounds and sensory characteristics of the fat substitute to explain the sensory properties of the reduced-fat product. It is interesting to note that dry characterized T5 despite it being one of the treatments with the lowest cooking losses and diameter reduction. In fact, due to the decrease in texture parameters in the burgers with *pijuayo* flour and the higher cooking yield, it would be expected that the reformulated treatments would be sensorially characterized as *tender* and *juicy*, but the physicochemical and sensory parameters were not related. This result agrees with Saldaña et al. [31], who observed that the results of descriptive analysis of texture and texture profile analysis of traditional and light mortadella were not correlated. This fact further complicates the decision criteria for recommending an appropriate replacement level and the type of by-product to replace fat in burgers; thus, viewing the results globally can show the overall picture of the product characteristics associated with its acceptance. However, since acceptance depends on consumer preferences, it is necessary to explore the preference profile of consumers to ensure that multivariate relationships of instrumental and sensory characteristics with sensory acceptance are representative. In this framework, an internal preference map and a dendrogram to identify consumer segments (clusters) were constructed.

According to the internal preference map (Figure 6a), it is suggested that pork fat can be partially replaced with *pijuayo* pulp and peel flour. This is because most consumers preferred the T1, T2, T3, and T4 treatments, which had a high density of liking vectors. However, the T5 treatment had a low density of liking vectors, indicating low consumer preference. These two tendencies are also reflected in the dendrogram (Figure 6b), which shows two clusters of consumers. The first cluster consists of consumers who preferred T1, T2, T3, and T4 burgers, while the second cluster, a minority, preferred T5. The two clusters represented 89% and 11% of the consumers, respectively. An evaluation of the responses of consumers in cluster 2 reveals that they tended to characterize T5 as *seasoned* (evaluated from Appendix A); therefore, this characteristic may have been important for the preference of these consumers. Therefore, the associations of physicochemical and sensory characteristics with acceptance through MFA can be justified because the preferences had consensus for most consumers.

### 3.3. Multiple Factor Analysis (MFA)

In the MFA, represented by 84.22% of the data variability, the samples were represented in three groups (Figure 7a): control (T1), burgers with 25% fat replaced by *pijuayo* flour (T2 and T4), burgers with 50% fat replaced by *pijuayo* pulp flour (T3), and burgers with 50% fat replaced by *pijuayo* peel flour. The distribution of treatments in the MFA was similar to that of the PCA, but certainly different from that of the CA. These differences are expected because of the different methodologies to express multivariate associations, but the results are complementary. Both physicochemical and sensory characteristics explain the differences between treatments in the MFA. Only the physicochemical and sensory characteristics that caused differences between the treatments were included, that is, those with relevance in the partial fat reduction by *pijuayo* flour in burgers. In addition, overall liking (in blue) was added as a supplementary variable in the MFA (Figure 7b). As shown in Figure 7b, the distribution of physicochemical characteristics (in red) was similar to those observed in the PCA, where b* and carbohydrates were in opposite positions to the other physicochemical parameters. However, in the sensory characteristics, the attribute grilled was located in the same quadrant as dry, but in the PA, grilled and dry had positive and negative impacts on overall liking, respectively.

The T1 treatment was characterized mainly by the instrumental characteristics, without associations with sensory characteristics. Similarly, T3 stood out for the physicochemical characteristics b* and carbohydrates. Physicochemical characteristics did not have a strong positive or negative relationship with overall liking; therefore, deciding the best product considering instrumental responses should be carefully considered. The samples with the lower fat reduction (T2 and T4) and inclusion of *pijuayo* flour stood out in the sensory attributes that positively drove liking in the PA. Within these attributes, tender and tasty were strongly correlated with overall liking. Thus, these associations suggest that replacing 25% fat with *pijuayo* pulp or peel flour improves the product’s tenderness and flavor. T5 was characterized mainly as dry and grilled, but these characteristics were opposite to overall liking in the MFA. Previously, dry was considered a negative driver of liking [14]. On the other hand, grilled was related to terms related to the cooking, such as fried, roasted, cooked, and toasted [20]. Thus, it is likely that consumers perceived the samples with *pijuayo* peel flour and 50% fat reduction to cause a requirement for excessive cooking of the product.

In short, replacing 25% pork backfat with *pijuayo* pulp and peel flour in burgers increases the acceptance of the product because they improve the tenderness and flavor of the product. For tenderness, beyond the low correlation between the physicochemical characteristics with overall liking in the MFA, the contribution that instrumental texture and yield properties could have with the sensory attribute tender should not be discarded, especially at the levels reached for T2 and T4, since these results could be a guide to optimize the acceptance of reduced-fat burgers added with fat substitute flour. Finally, *pijuayo* fruit could be a strong candidate for flavor enhancer of reduced-fat meat products.

## 4. Conclusions

The effect of *pijuayo* pulp and peel flours as animal fat replacers in burgers was evaluated on the physicochemical and sensory characteristics by multivariate statistical analysis. The flours affected the physicochemical characteristics observed by PCA. At the higher level of fat replacement (50%), the burgers were characterized by a high carbohydrate content and instrumental yellowness. In the sensory characteristics evaluated by CA, the control treatment and the 25% fat replacement were positive to highlight sensory attributes with a positive impact on liking. However, overall liking was higher for treatments with 25% fat reduction, even higher than the control. The internal preference map showed that consumers drove their preference to the control and fat-reduced treatments, except the treatment with 50% fat replacement using *pijuayo* peel flour, of which only a small segment of consumers (11% of the total) preferred that treatment. The MFA showed that the sensory characteristics tender and tasty were strongly correlated with overall liking, demonstrating that *pijuayo* pulp or peel flours at 25% fat replacement improved the tenderness and flavor of fat-reduced burgers. The development of meat products with a healthier profile is a real public health need, and Amazon fruits have the potential to improve the image of this type of products. The industrial use of *pijuayo* as an ingredient in meat products could be an economic support for Amazon settlers. Future studies should evaluate the optimization of fat replacement to explore other inclusion levels of *pijuayo* flour between 25% and 50% of fat replacement, to explore specific antioxidant compounds in the *pijuayo* pulp and peel flour, to evaluate the sensory characteristics and flavor-enhancing compounds of the *pijuayo* fruit to explain the sensory properties of the burgers, to evaluate nutritional aspects of the inclusion of *pijuayo* such the fatty acid profile, as well as pilot-scale evaluations as a requirement for technology transfer to industry.

## Figures and Tables

**Figure 1 foods-13-01619-f001:**
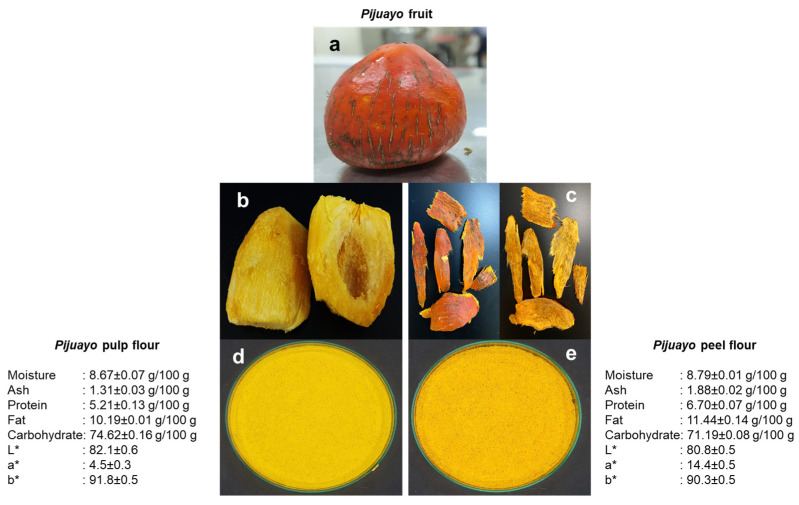
*Pijuayo* fruit processing: (**a**) *pijuayo* fruit; (**b**) cooked *pijuayo* pulp; (**c**) external (left side) and internal (right side) parts of the cooked *pijuayo* peel; (**d**) *pijuayo* pulp flour; and (**e**) *pijuayo* peel flour. The proximal composition and instrumental color (L*, a*, b*) of the *pijuayo* flours are also shown.

**Figure 2 foods-13-01619-f002:**
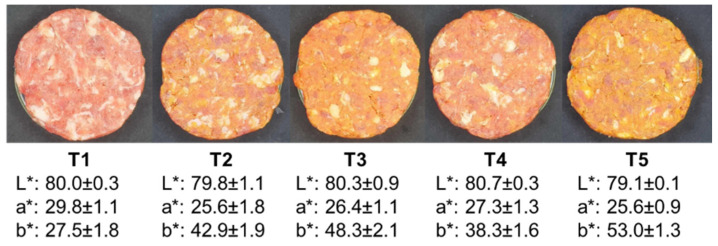
Photographs of the burger treatments. Pork fat substitution with 0% (T1), 25% (T2), and 50% (T3) *pijuayo* pulp flour, and 25% (T4) and 50% (T5) *pijuayo* peel flour. At the bottom of each treatment, the average values and standard deviation of the instrumental color (L*, a*, b*) are shown.

**Figure 3 foods-13-01619-f003:**
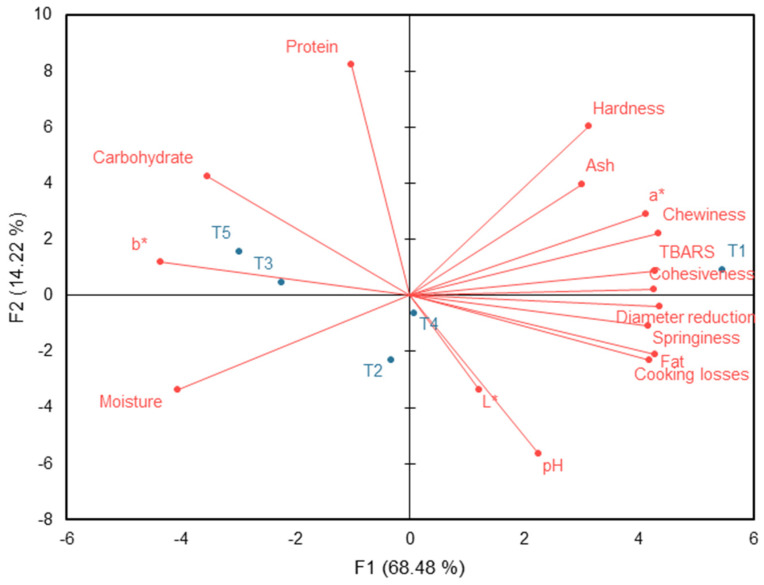
Principal Component Analysis (PCA) of physicochemical characteristics of burger treatments. Pork fat substitution with 0% (T1), 25% (T2), and 50% (T3) *pijuayo* pulp flour, and 25% (T4) and 50% (T5) *pijuayo* peel flour.

**Figure 4 foods-13-01619-f004:**
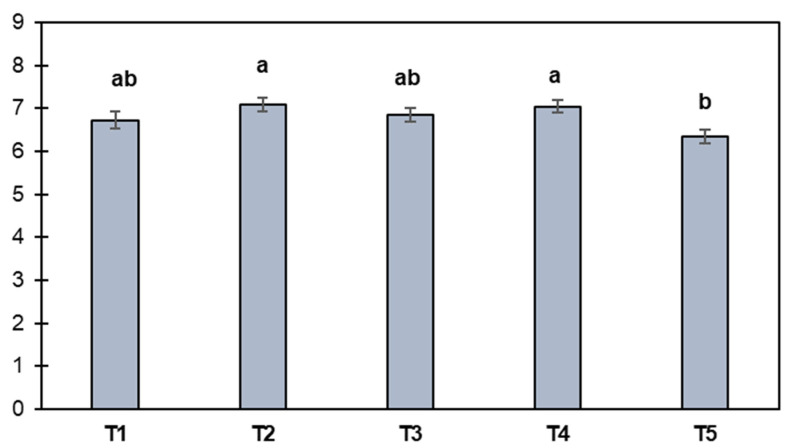
Overall liking of burger treatments (mean and standard error of the mean). Pork fat substitution with 0% (T1), 25% (T2), and 50% (T3) *pijuayo* pulp flour, and 25% (T4) and 50% (T5) *pijuayo* peel flour. According to Tukey’s test, different letters on the bars represent a significant difference (*p* < 0.05) between burger treatments.

**Figure 5 foods-13-01619-f005:**
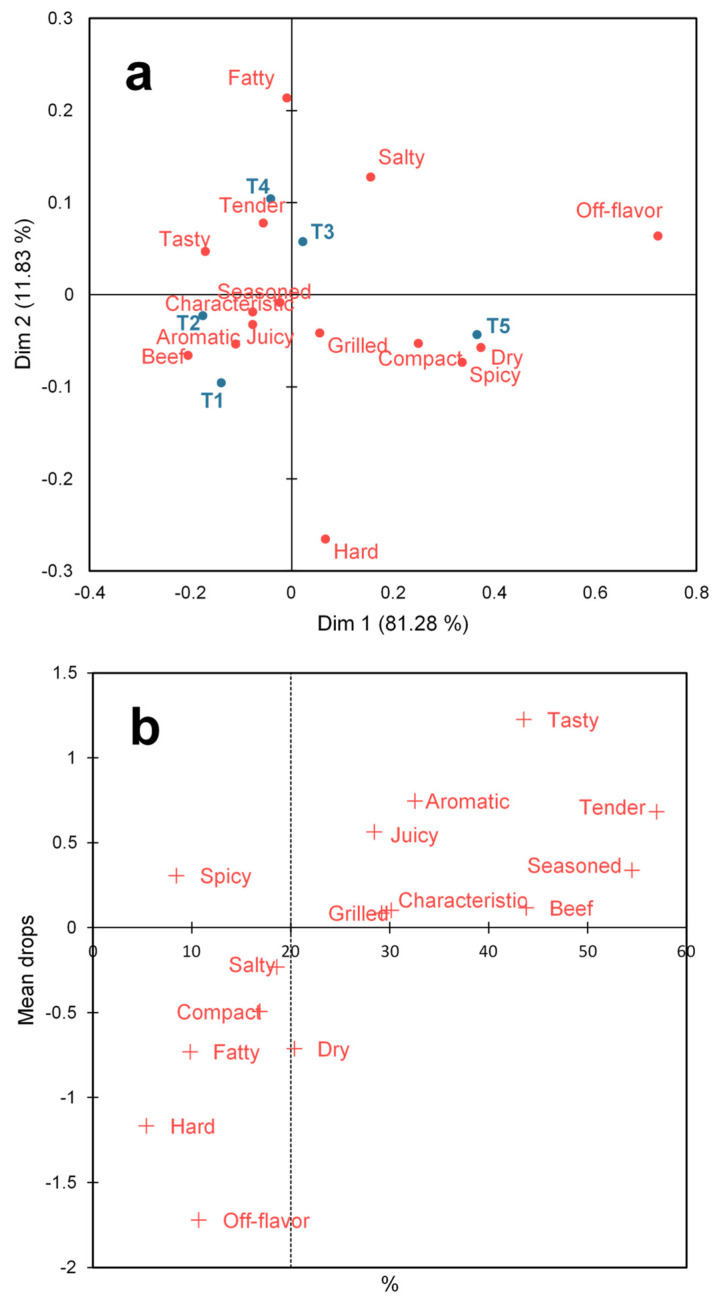
Correspondence Analysis (CA) (**a**) and Penalty Analysis (PA) (**b**) of sensory characteristics of burger treatments. Pork fat substitution with 0% (T1), 25% (T2), and 50% (T3) *pijuayo* pulp flour, and 25% (T4) and 50% (T5) *pijuayo* peel flour.

**Figure 6 foods-13-01619-f006:**
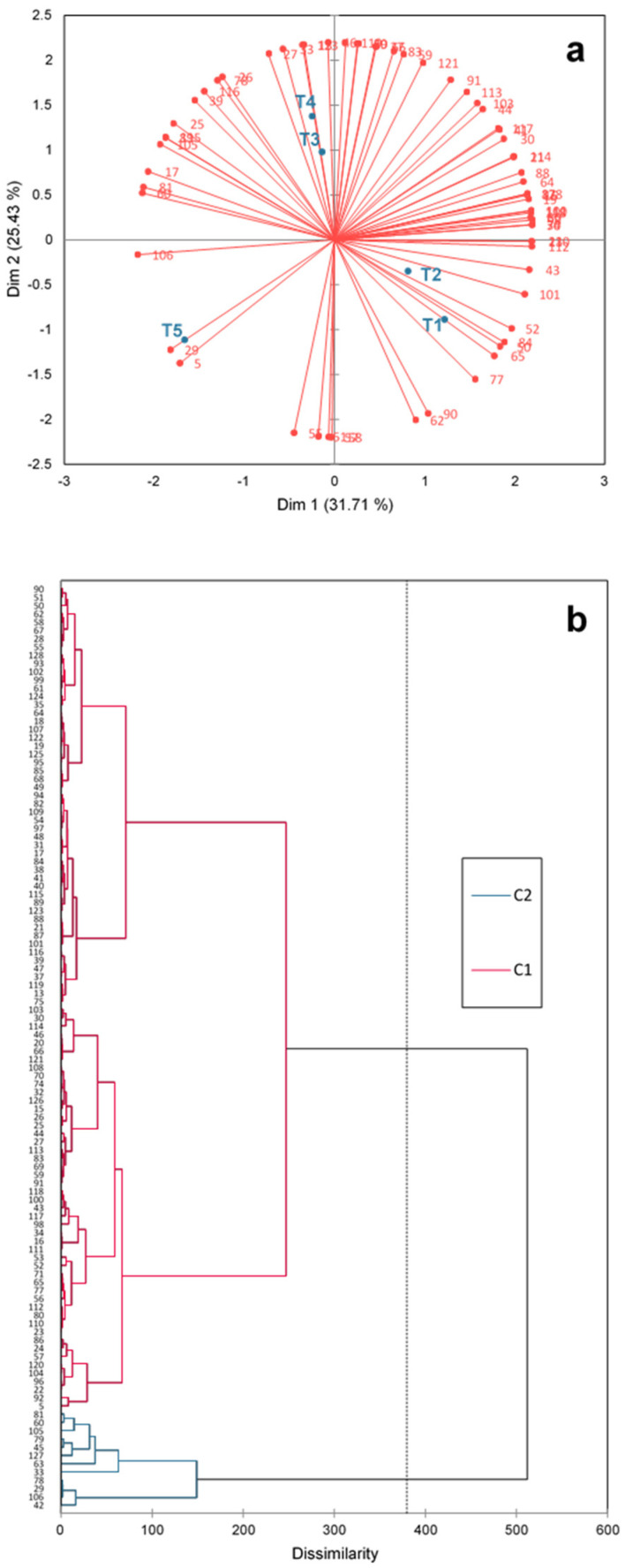
Internal preference map (**a**) and dendrogram (**b**) of consumer preferences on burgers treatments. Pork fat substitution with 0% (T1), 25% (T2), and 50% (T3) *pijuayo* pulp flour, and 25% (T4) and 50% (T5) *pijuayo* peel flour. C1: cluster 1; C2: cluster 2.

**Figure 7 foods-13-01619-f007:**
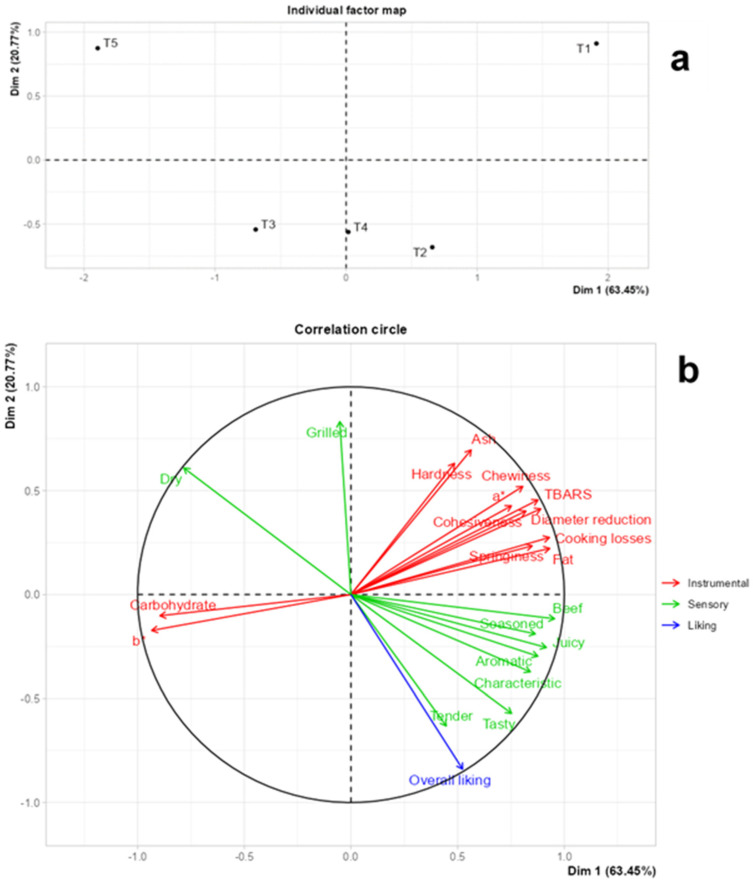
Multiple Factor Analysis (MFA) of burger treatments (**a**) and physicochemical characteristics (instrumental), sensory characteristics and overall liking (**b**). Pork fat substitution with 0% (T1), 25% (T2), and 50% (T3) *pijuayo* pulp flour, and 25% (T4) and 50% (T5) *pijuayo* peel flour.

## Data Availability

The original contributions presented in the study are included in the article, further inquiries can be directed to the corresponding author.

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
