# Peer review of "Exploring Pijuayo (Bactris gasipaes) Pulp and Peel Flours as Fat Replacers in Burgers: A Multivariate Study on Physicochemical and Sensory Traits"

_foods, 2024, doi:10.3390/foods13111619_

Round 1

Reviewer 1 Report

Comments and Suggestions for Authors

Manuscript ID: foods-3008554

Title: Exploring Pijuayo (Bactris gasipaes) Pulp and Peel Flours as Fat Replacers in Burgers: A Multivariate Study on Physicochemical and Sensory Traits

The study assesses the sensory and physicochemical outcomes of the partial replacement of animal fat in beef-based burgers with pijuayo pulp and peel flours and, as such, fits the journal scope. The manuscript is relevant for the field, clear, presented in a well-structured manner and written in an appropriate way. The authors address well-defined questions (possibility of partial replacement of animal fat in beef-based burgers with pijuayo pulp and peel flours and the impact of such a procedure on the selected characteristics of the final product) utilizing appropriate and technically sound approach (instrumental analysis, sensory analysis, statistical tools). Informed consent was obtained from all the subjects involved in the sensory assessment. The methods and software are described with sufficient details. According to the Data Availability Statement, inquiries related to data not presented in the manuscript can be directed to the corresponding author. The data is robust enough to draw conclusions. The comprehensive statistical assessment is based on Principal Component Analysis, Correspondence Analysis and Multiple Factor Analysis. The statistical analyses and statistical reporting are appropriate and well described. The results are significant and interpreted appropriately and consistently throughout the manuscript, providing an advancement of the current knowledge. The authors clearly emphasized the strengths of the study and provided context for their results by comparing them with relevant previously published studies. The conclusions are justified and consistent with the evidence and arguments presented. The conclusions are interesting for a wide readership of the journal, and thus, there is an overall benefit to publishing this work.

Regarding the cited references, they are mostly recent publications and relevant. However, a significant number of self-citations has to be noted. The authors should consider to omit some of self-citations that are not necessary or not directly related to the topic of the current manuscript (for example, references 15 and 38).

Minor comments:

Line 222: as only two levels of substitution were applied, terms higher and lower seem more appropriate than highest and lowest.

Line 285: there is a typographical error in linking – it should be liking.

Line 548: the b) mark is missing from the title of Figure 7.

Line 562, Conclusion: together with the experimentally determined facts, in the conclusion it is desirable to emphasize the importance of the conducted research and the obtained results as well as their applicability, bearing in mind the economic, public health and other relevant aspects.

Author Response

The study assesses the sensory and physicochemical outcomes of the partial replacement of animal fat in beef-based burgers with pijuayo pulp and peel flours and, as such, fits the journal scope. The manuscript is relevant for the field, clear, presented in a well-structured manner and written in an appropriate way. The authors address well-defined questions (possibility of partial replacement of animal fat in beef-based burgers with pijuayo pulp and peel flours and the impact of such a procedure on the selected characteristics of the final product) utilizing appropriate and technically sound approach (instrumental analysis, sensory analysis, statistical tools). Informed consent was obtained from all the subjects involved in the sensory assessment. The methods and software are described with sufficient details. According to the Data Availability Statement, inquiries related to data not presented in the manuscript can be directed to the corresponding author. The data is robust enough to draw conclusions. The comprehensive statistical assessment is based on Principal Component Analysis, Correspondence Analysis and Multiple Factor Analysis. The statistical analyses and statistical reporting are appropriate and well described. The results are significant and interpreted appropriately and consistently throughout the manuscript, providing an advancement of the current knowledge. The authors clearly emphasized the strengths of the study and provided context for their results by comparing them with relevant previously published studies. The conclusions are justified and consistent with the evidence and arguments presented. The conclusions are interesting for a wide readership of the journal, and thus, there is an overall benefit to publishing this work.

Regarding the cited references, they are mostly recent publications and relevant. However, a significant number of self-citations has to be noted. The authors should consider to omit some of self-citations that are not necessary or not directly related to the topic of the current manuscript (for example, references 15 and 38).

Thank you very much for your comments. As requested, the following references were removed:

Saldaña, E.; Saldarriaga, L.; Cabrera, J.; Siche, R.; Behrens, J.H.; Selani, M.M.; de Almeida, M.A.; Silva, L.D.; Silva Pinto, J.S.; Contreras-Castillo, C.J. Relationship between Volatile Compounds and Consumer-Based Sensory Characteristics of Bacon Smoked with Different Brazilian Woods. Food Res. Int. 2019, 119, 839–849, doi:https://doi.org/10.1016/j.foodres.2018.10.067.

Saavedra, A.R.L.; Rios-Mera, J.D.; Imán, A.; Vásquez, J.; Saldaña, E.; Siche, R.; Tello, F. A Sequential Approach to Reduce Sodium Chloride in Freshwater Fish Burgers Considering Chemical, Texture, and Consumer Sensory Responses. LWT 2022, 167, 113854, doi:10.1016/J.LWT.2022.113854.

Iman, A.; Rios-Mera, J.D.; Rengifo, E.; Palomino, F.; Vela-Paredes, R.; Vásquez, J.; García de Sotero, D.E.; Saldaña, E.; Siche, R.; Tello, F. A Comparative Study of Freshwater Fish Burgers Made from Three Amazonian Species: Omega 3 Fortification and Sodium Reduction. Foods 2024, 13.

However, reference 38 from the previous version of the manuscript was not eliminated, because it was important for the discussion of the results, located between lines 455–462 of this new version.

Minor comments:

Line 222: as only two levels of substitution were applied, terms higher and lower seem more appropriate than highest and lowest.

It was corrected as requested by the reviewer (line 247, 248, 255, 389, 391, 550, 610).

Line 285: there is a typographical error in linking – it should be liking.

We apologize for the mistake. It was corrected (line 311).

Line 548: the b) mark is missing from the title of Figure 7.

Corrected.

Line 562, Conclusion: together with the experimentally determined facts, in the conclusion it is desirable to emphasize the importance of the conducted research and the obtained results as well as their applicability, bearing in mind the economic, public health and other relevant aspects.

According to the reviewer's comments, the conclusions were improved, suggesting the applications of the results and the challenges to be faced in future studies (lines 620-630). This information was also briefly mentioned in the abstract of the manuscript (lines 40-45).

Reviewer 2 Report

Comments and Suggestions for Authors

Exploring Pijuayo (Bactris gasipaes) Pulp and Peel Flours as Fat Replacers in Burgers: A Multivariate Study on Physicochemical and Sensory Traits

Abstract

  1.      In the first line of abstract author mentioned that pijuayo pulp and peel flours were used as fat replacer kindly care to explain why? Write one or two lines about the lipid profile of the pijuayo pulp and peel flours. So it will be easy for the readers to understand the use of pijuayo pulp and peel flours as a fat replacer.

  2.      It would be helpful to briefly mention why reducing animal fat in burgers is significant. Is it for health reasons, environmental concerns or another purpose? This context can help readers understand the relevance and potential impact of the study.

  3.      In the line 31 and 32 it is mentioned that a small segment of consumers preferred burgers with greater fat replacement, despite lower overall liking. It would be beneficial to explore the reasons behind this preference, as it could offer insights into consumer attitudes toward healthier or alternative food options.

  4.      The abstract could briefly discuss the practical implications of the findings and suggest potential avenues for future research. For example, could pijuayo flour be further optimized to enhance both health benefits and sensory appeal in burgers?

Introduction

  5.      Author didn’t mention regarding the harvesting and cultivation of pijuayo like in which region it grow how much cultivated as well as how much it cost. Write two or three lines about this information to make this fruit more interesting

  6.      While the excerpt mentions the potential benefits of using pijuayo as a fat replacement, it could provide more specific information about its application in meat products like burgers. For example, does pijuayo flour alter the sensory attributes or cooking properties of burgers, and if so, how?

  7.      It would be beneficial to include information about any potential challenges or limitations associated with using pijuayo or other botanical resources as fat substitutes. This could include issues related to flavor, texture, or consumer acceptance.

Results

  8.      In the line 222 and 223 author mentioned that the burgers with the highest inclusion of pijuayo flour (T3 and T5), burgers with the lowest 222 inclusion of pijuayo flour (T2 and T4) and the control (T1) in it kindly mentioned the specific amount of the inclusion of the pijuayo in the T3, T5, T2, T4 of the burger.

  9.      While the paragraph of sensory analysis is informative, it could benefit from breaking down some of the longer sentences into shorter, more digestible segments. This will enhance readability and make it easier for readers to follow the flow of information.

  10.  In the sensory analysis when discussing the increases in acceptability attributed to the characteristics of fat substitutes, it would be helpful to clearly state how these characteristics contribute to improved liking scores. This could provide readers with a clearer understanding of the mechanisms at play.

Conclusion

  11.  Including a brief discussion of any limitations of the study such as sample size, specific characteristics of the burgers, or potential confounding factors, would provide a balanced perspective. Additionally, suggesting avenues for future research based on the study's findings could add value to the discussion.

Author Response

Abstract

  1. In the first line of abstract author mentioned that pijuayo pulp and peel flours were used as fat replacer kindly care to explain why? Write one or two lines about the lipid profile of the pijuayo pulp and peel flours. So it will be easy for the readers to understand the use of pijuayo pulp and peel flours as a fat replacer.

As requested by the reviewer, the reason for using pijuayo as a fat replacer in burgers was described (lines 23-27).

  1. It would be helpful to briefly mention why reducing animal fat in burgers is significant. Is it for health reasons, environmental concerns or another purpose? This context can help readers understand the relevance and potential impact of the study.

The fat reduction is for health reasons, described on lines 23-25.

  1. In the line 31 and 32 it is mentioned that a small segment of consumers preferred burgers with greater fat replacement, despite lower overall liking. It would be beneficial to explore the reasons behind this preference, as it could offer insights into consumer attitudes toward healthier or alternative food options.

Thank you very much for the observation that helped to better explore the consumer responses who preferred the treatment with greater inclusion of pijuayo peel flour. In this sense, the main reason is explained in line 37 of the abstract and in lines 478-481 of results section.

  1. The abstract could briefly discuss the practical implications of the findings and suggest potential avenues for future research. For example, could pijuayo flour be further optimized to enhance both health benefits and sensory appeal in burgers?

According to the reviewer suggestion, the final part of the abstract was improved (lines 40-45).

Introduction

  1. Author didn’t mention regarding the harvesting and cultivation of pijuayo like in which region it grow how much cultivated as well as how much it cost. Write two or three lines about this information to make this fruit more interesting.

As requested, information on harvesting of pijuayo was added (lines 88-91). About the cost, no information was found, but we believe that it can be variable according to the country where is consumed.

  1. While the excerpt mentions the potential benefits of using pijuayo as a fat replacement, it could provide more specific information about its application in meat products like burgers. For example, does pijuayo flour alter the sensory attributes or cooking properties of burgers, and if so, how?

In lines 83-86 we cite our previous work on the effect of pijuayo flour on the physicochemical properties of burgers, which included cooking properties. We indicate that pijuayo alters the physicochemical composition, but it is necessary to explore whether these changes such as the reduction of texture properties are sensorially acceptable, in addition to exploring the sensory properties and acceptance of the burgers. Regarding how pijuayo flour alters the burger properties, this aspect is part of the discussion of this work, so it cannot yet be described in the introduction. Therefore, we consider that description in the aforementioned lines responds to the reviewer's comment.

  1. It would be beneficial to include information about any potential challenges or limitations associated with using pijuayo or other botanical resources as fat substitutes. This could include issues related to flavor, texture, or consumer acceptance.

Information was added in lines 88-91 according to the reviewer’s comment.

Results

  1. In the line 222 and 223 author mentioned that the burgers with the highest inclusion of pijuayo flour (T3 and T5), burgers with the lowest inclusion of pijuayo flour (T2 and T4) and the control (T1) in it kindly mentioned the specific amount of the inclusion of the pijuayo in the T3, T5, T2, T4 of the burger.

To better understanding, these lines were modified, where the percentages of fat replacement were included (lines 247-249).

  1. While the paragraph of sensory analysis is informative, it could benefit from breaking down some of the longer sentences into shorter, more digestible segments. This will enhance readability and make it easier for readers to follow the flow of information.

The writing was improved, especially in the description and discussion of the overall liking results (lines 312-313; 326-337; 345-350).

  1. In the sensory analysis when discussing the increases in acceptability attributed to the characteristics of fat substitutes, it would be helpful to clearly state how these characteristics contribute to improved liking scores. This could provide readers with a clearer understanding of the mechanisms at play.

In line with the previous observation, the description of overall liking results and discussion was improved. In short, it is indicated that the sensory properties and composition of the pijuayo fruit and the effect on the reduction of texture parameters could be involved in the increase in the acceptance of burgers (lines 326-338, 345-350).

Conclusion

  1. Including a brief discussion of any limitations of the study such as sample size, specific characteristics of the burgers, or potential confounding factors, would provide a balanced perspective. Additionally, suggesting avenues for future research based on the study's findings could add value to the discussion.

Thank you very much for your comments. The conclusion was improved as the reviewer suggested (lines 620-630).

Reviewer 3 Report

Comments and Suggestions for Authors

This work evaluated the effect of pijuayo pulp and peel flours as animal fat replacers in burgers on the physicochemical and sensory characteristics by PCA, CA and MFA. The research is interesting, and the results might help to manufacture low animal fat burgers. There are several points that need to be improved.

1.       The original data of the physicochemical determination and sensory analysis should be provided.

2.       The contents of phenolic compounds or carotenoids which might correlate with decreased lipid oxidation can be determined.

3.       The methods can be described more detailed to give a general understanding about the methods without check the references.

4.       The limitation of this research should be discussed. For example, the results suggested that 25% fat reduction show better effects. However, this experiment only performed three treatments (0, 25% and 50% fat reduction). The differences of these treatments are great. What about the effects of little lower or higher than this (such as 15% or 35% fat reduction)?

Author Response

This work evaluated the effect of pijuayo pulp and peel flours as animal fat replacers in burgers on the physicochemical and sensory characteristics by PCA, CA and MFA. The research is interesting, and the results might help to manufacture low animal fat burgers. There are several points that need to be improved.

  1. The original data of the physicochemical determination and sensory analysis should be provided.

Thank you very much for the valuable comments of the reviewer. Physicochemical and sensory raw data was included as supplementary material in this new version.

  1. The contents of phenolic compounds or carotenoids which might correlate with decreased lipid oxidation can be determined.

Unfortunately, phenolic and carotenoid compounds were not explored, which is a limitation of this study. However, we indicate in the abstract and conclusion that these components should be explored in future studies.

  1. The methods can be described more detailed to give a general understanding about the methods without check the references.

The methods were described in details, such as the proximal composition of the flours (lines 135-142) and the ingredients and their concentrations used in the production of the burgers (lines 158-163).

  1. The limitation of this research should be discussed. For example, the results suggested that 25% fat reduction show better effects. However, this experiment only performed three treatments (0, 25% and 50% fat reduction). The differences of these treatments are great. What about the effects of little lower or higher than this (such as 15% or 35% fat reduction)?

Thank you very much for your comments. The conclusion was improved as the reviewer suggested.

Round 2

Reviewer 3 Report

Comments and Suggestions for Authors

The authors have improved the manuscript accoring to the comments. I think it could be accepted for publication with some editorial changes.